# Modulation of Lower-Limb Muscle Activity in Maintaining Unipedal Balance According to Surface Stability, Sway Direction, and Leg Dominance

**DOI:** 10.3390/sports10100155

**Published:** 2022-10-17

**Authors:** Arunee Promsri

**Affiliations:** 1Department of Physical Therapy, School of Allied Health Sciences, University of Phayao, Phayao 56000, Thailand; arunee.pr@up.ac.th; Tel.: +66-54-466-666 (ext. 3817); 2Department of Sport Science, University of Innsbruck, 6020 Innsbruck, Austria; 3Unit of Excellence in Neuromechanics, School of Allied Health Sciences, University of Phayao, Phayao 56000, Thailand

**Keywords:** neuromuscular control, postural control, single-leg stance, electromyography (EMG), center of pressure (COP)

## Abstract

Determining temporal similarity in shape between electromyographic (EMG) and center-of-pressure (COP) signals reflects neuromuscular control in terms of which relevant muscles are involved in maintaining balance. The current study aimed to investigate a cross-correlation between seven lower-limb EMG activities and COP displacements, simultaneously measured in 25 young adults unipedally balancing on stable and multiaxial-unstable surfaces. The effect of surface stability, sway direction, and leg dominance was then tested on two EMG–COP correlation levels: individual muscles and groups (patterns) of multi-muscles involved in postural sway, as determined by principal component analysis (PCA). The results show that two factors demonstrate their effects only at the level of individual muscles: sway direction (*p* ≤ 0.003) and leg dominance (*p* = 0.003). Specifically, the semitendinosus, tibialis anterior, peroneus longus, and soleus correlate more with the mediolateral postural sway than with the anteroposterior postural sway, except for the gastrocnemius medialis. Additionally, balancing on the non-dominant leg shows a lower correlation between the semitendinosus and postural sway than on the dominant leg. The current findings suggest that when achieving unipedal equilibrium, the postural control system may be constrained the most in the specific muscles of the least steady conditions, e.g., the frontal plane and non-preferred leg.

## 1. Introduction

An ability to effectively control the whole-body posture on one leg is essential for completing various daily functional motor tasks (e.g., walking, turning, dressing, and climbing up and down stairs) and participating in physical activities or sports (e.g., jumping and running). In this sense, according to a small base of support in the mediolateral direction that restricts control of postural stability, a single-leg stance has been considered one of the clinical balance tests in assessing postural control [1,2,3], predicting the risk of falling [4,5,6], and determining the laterality effect on postural control [7,8,9]. In addition, this standing posture is also used in training to improve neuromuscular control, regain postural balance, and enhance physical or athletic performance [2,3,10,11].

In order to assess postural control, the most popular method is to examine motor behavior, typically seen as body (postural) sway, by measuring the center-of-pressure (COP) displacements received from the force plate [12]. Posturographic information obtained on this basis reflects the outcome of the ground reaction force and moment, indirectly measuring human postural control via COP-based variables, e.g., sway area, sway path length, and sway velocity [12]. However, monitoring two-dimensional COP movements alone provides insufficient information about neuromuscular control in terms of which postural muscles are involved in generating postural sway (i.e., driving COP motions) [13]. In other words, the mechanisms controlling human posture have been suggested to be examined directly via three-dimensional postural movements and muscle activations rather than indirectly via sensorimotor system-influenced variables like COP-based variables [13], since the neuromuscular system controls posture via muscles that produce relative movements across the body segments [14]. Hence, to shed light on the neuromuscular mechanisms underlying unipedal postural sway, examining temporal similarity (i.e., form) between electromyographic (EMG) activity and COP signals can describe an overview of EMG–COP correlation [15,16], revealing the relevant muscles engaged in maintaining equilibrium (i.e., generating postural sway) [16,17,18,19,20,21].

The current study aimed to better understand the neuromuscular control of unipedal standing posture by examining the extent to which lower-limb EMG activities are correlated with COP motions during balancing on stable and multiaxial-unstable surfaces, of which three factors that might influence postural control—surface stability, sway direction, and leg dominance—were the focus areas. With a less stable surface and a small base of support in the mediolateral direction, postural stability is likely more constrained than with a more stable surface and a larger support area in the anteroposterior direction, representing the distinct role of individual muscles in maintaining balance under specific conditions [19,20]. Additionally, the preferential use of one leg over another for unimanual lower-limb motor tasks (i.e., leg dominance) [22] influences asymmetry in sensorimotor control between the two legs [7,8], confirming its effects as an internal risk factor for sports-related lower-limb injuries [23,24,25]. These three factors were tested on two EMG–COP correlation levels: individual muscles and groups (patterns) of multi-muscles involved in postural sway. First, since individual muscles play an essential and distinct role according to their anatomical functions [26], it was hypothesized that at the level of individual muscles, the tested factor effects would be observed in the specific muscles relevant to the specific conditions. Second, given the numerous degrees of freedom of the motor apparatus, the number of variables (i.e., muscles) that must be controlled can be reduced by organizing the inherent redundancies into synergies (i.e., patterns) [27]. Hence, if most joints/movements are controlled by a specific group of multi-muscles (i.e., task-dependent muscle synergies) [28,29,30], it was hypothesized that the tested factor effects would manifest in the specific multi-muscle patterns relevant to the specific conditions.

In summary, the link between lower-limb EMG activity and COP trajectory was investigated in physically active young adults performing unipedal balancing on stable and multiaxial-unstable surfaces. Three factors—surface stability, sway direction, and leg dominance—were tested on two EMG–COP correlation levels: individual muscles and multi-muscle patterns. The tested factor effects were expected to be observed in the specific muscles or in the specific multi-muscle patterns relevant to the current task. Knowledge of the inherent neuromuscular control of unipedal balance may benefit sports-related injury prevention and rehabilitation.

## 2. Materials and Methods

### 2.1. Participants

Twenty-five physically active young adults (14 males and 11 females) with no neurological or musculoskeletal problems and no balance-specific training in the previous six months participated in the current study. Most volunteers (96%) indicated their right leg as the dominant leg. Kicking a ball was used to define the dominant leg, since this task had proven more effective in assessing inter-limb differences in unipedal postural control than using a single-leg stance test to determine leg dominance [7]. The Board for Ethical Questions in Science at the University of Innsbruck, Austria, had approved the study protocol, and all participants signed a written informed consent before their participation. The characteristics of the participants are represented in Table 1.

### 2.2. Equipment

Surface electromyography (EMG) with a sampling rate of 1500 Hz (Noraxon TeleMyo^TM^ 2400T G2 Direct Transmission System, Noraxon Inc., Scottsdale, AZ, USA) was used to measure the EMG activity (μV) of seven muscles in each leg: the rectus femoris (RF), semitendinosus (ST), biceps femoris (BF), tibialis anterior (TA), peroneus longus (PL), gastrocnemius medialis (GM), and soleus (SO). After shaving, scrubbing, and cleaning the skin, the disposable pre-gelled bipolar 22-mm Ag/AgCl surface adhesive round-electrodes (Ambu Neuroline 720 01-K/12; Ambu, Ballerup, Denmark) with a 20-mm inter-electrode center-to-center distance were applied over the lower-leg muscles according to the SENIAM guidelines [31]. A low-resistance impedance (<6 kΩ) was obtained between the electrodes and the skin. The reference electrode was placed on the tibial tuberosity of the right leg. The EMG cables were taped to the skin to reduce movement artifacts.

The center-of-pressure (COP) displacements (mm) were measured using a force plate (OPTIMA^TM^ AMTI force plate, AMTI, Watertown, NY, USA) with a sampling rate of 1250 Hz. An MFT Challenge Disc (Trend Sport Trading GmbH., Austria) was chosen to induce an adjusted postural movement that responded to surface instability. This balance board consists of an upper 44-cm diameter round plate connected to a base circle plate by a group of four-rubber cylinders with an 8-cm height in the middle of the plates. The force plate was synchronized with the EMG system and was operated through Nexus 2.2.3 software (Vicon Motion Systems Ltd., Oxford, UK).

### 2.3. Experimental Procedures

All participants randomly performed an 80-s unipedally barefooted stand for each support surface on each foot. In order to standardize the starting position [7], all volunteers were asked to place their hands on their hips, flex the hip and knee of the lifted leg at 20 and 45 degrees, respectively, and place the marked point (base of the second metatarsal bone) on each stance-foot over the center of the reticle crossline marked on the center of the force plate, and align the second toe with the anteroposterior crossline for stable-surface conditions or to the balance board crossline for unstable-surface conditions (Figure 1A).

For unstable-surface conditions, the center of the balance board was placed over the center of the reticle crossline marked on the force plate, to which the marked anteroposterior and mediolateral diameters taped onto the boards were aligned to these crossed lines to standardize the position of the balance boards [32]. Hence, the same fulcrum position was set for all trials of all participants.

During testing, participants were required to look straight ahead at a target (a 10-cm diameter red circle on a white background) set at the individual’s eye level on a wall approximately five meters away, stand still for stable conditions, keep the balancing board horizontal for unstable conditions, and rest for one to three minutes after each trial [7]. During the resting time, participants could sit or stand, but they were not allowed to stand on the balance board during the rest time.

### 2.4. Data Analysis

#### 2.4.1. EMG and COP Data Pre-Processing

MATLAB^TM^ version 2020a was used for all data processing (MathWorks Inc., Natick, MA, USA). In order to remove movement artifacts and high-frequency noise, individual EMG signals were filtered with the second-order band-pass Butterworth filter at 20–500 Hz and then rectified [33]. Anteroposterior (*ap*) and mediolateral (*ml*) COP displacements (Figure 1B) were estimated from the ground reaction forces (Fx and Fy) and moments of forces (Mx and My) with consideration of the height (h) of the base of support above the force plate [12].
(1)$$\mathrm{COP}ap=\left(-h*Fx-My\right)/Fz$$
(2)$$\mathrm{COP}ml=\left(-h*Fy+Mx\right)/Fz$$

The EMG and COP signals were then smoothed with the same filtering with a third-order zero-phase 10-Hz low-pass Butterworth filter [14]. Both signals were then down-sampled to 250 Hz, and their middle 60-s data consisting of 15,000 timepoints (Figure 1C) were selected for further analysis [14].

#### 2.4.2. Calculating EMG–COP Correlation

Individual middle 60-s data of EMG and COP signals were separated into ten subsets, each with six seconds of data (1500 timepoints) [34]. A normalized cross-correlation analysis was applied to each subset to increase the reliability of determining the cross-correlation coefficient $\left(r\right)$ observed as a peak at the time delay $\left(\tau \right)$ in cross-correlation graphs (Figure 1D) [15,16,17], representing the amount of the EMG signal involved in the COP signal [15]. In each participant, cross-correlations were determined for the anteroposterior and mediolateral displacements. An absolute value of r (ǀ𝑟ǀ) between 0.1 and 0.3 has been classified as small, between 0.3 and 0.5 as moderate, and between 0.5 and 1.0 as a large correlation [35]. For further analysis, the average value of ǀ𝑟ǀ from all subsets of each volunteer was selected [16,34], and Fisher’s r-to-z transformation normalized the |r| before statistical comparison [36].

#### 2.4.3. Analyzing Patterns of EMG–COP Correlations

Since examining the cross-correlation between EMG activities and COP displacements reveals the shared content or similar form of these two signals [15], analyzing the patterns of EMG–COP correlations [34] reflects the cooperative involvement of multiple muscles in generating postural sway, i.e., driving COP motions. In this sense, a principal component analysis (PCA) [34] was used to explore the patterns of EMG–COP correlations by applying to the |r| of 200 vectors (2 [legs] × 2 [COP directions] × 2 [support surfaces] × 25 [participants]) that were normalized to z-score before calculation. PCA was calculated using a singular value decomposition algorithm, providing one set of eigenvectors (principal component, PC_k_), eigenvalues (EV_k_), and the scores common to all trials of all participants, where _k_ denotes the order of the eigenvector.

The relative explained variances (rVAR_k_) of the scores were then calculated as subject-specific variables that directly corresponded to the EV_k_, quantifying how much each PC_k_ contributed to this subject’s overall variance [7]. If differences exist in the rVAR_k_ between conditions, this indicates a difference in the correlated patterns of EMG–COP correlations in the sense that the participants’ overall variance influences specific PC_k_ [7].

### 2.5. Statistical Analysis

All statistical analyses were performed using IBM SPSS Statistics Software Version 26.0 (SPSS Inc., Chicago, IL, USA). The alpha value was set to a = 0.05. The Shapiro-Wilk tests suggest a repeated-measures ANOVA for testing the effects of surface stability, sway direction, and leg dominance. The Holm-Bonferroni correction [37] was used to manage the family-wise error rate and adjust the alpha level (seven comparisons for each factor). Therefore, the *p*-values smaller than *p* < 0.05 but not satisfied with the Holm-Bonferroni correction were interpreted as a statistical trend. Cohen’s *d* was computed as the effect size for the post hoc comparisons, with *d* = 0.2 considered a small effect size, *d* = 0.5 considered a medium effect size, and *d* = 0.8 considered a large effect size [35].

## 3. Results

### 3.1. An Overview of EMG–COP Correlation

All participants were able to stand in a single-leg position for the entire duration of all balancing trials without touching the floor with their lifted leg. Figure 2 illustrates the r-by-τ graph, representing an example of individual EMG–COP correlation drawn from the first subset of data. The alignments of median time delays $\left(\tau m\right)$ are observed in the specific muscles and in the specific COP direction. Specifically, in the AP direction, the alignments exist for the GM and SO in both stable and unstable surfaces, while in the ML direction, the alignments appear for the ST, TA, PL, GM, and SO in both stable and unstable surfaces.

### 3.2. Individual Pairs of EMG–COP Correlation

Table 2 represents the average values of the absolute z-transformed EMG–COP correlation coefficients (ǀ𝑟ǀ) tested by three factors: surface stability, sway direction, and leg dominance. From an overview, the average values of ǀ𝑟ǀ range between 0.3 and 0.7, demonstrating a moderate-to-large EMG–COP correlation that varies depending on specific muscles and variables. In addition, the highest value of ǀ𝑟ǀ is observed in the PL in the ML direction according to the sway direction effects.

The main results show that two factors—sway direction and leg dominance—only demonstrate their effects at the level of individual muscles, which are seen in the specific muscles. For the sway direction effects, the higher correlation coefficients are observed for the ST (*p* < 0.001), TA (*p* < 0.001), PL (*p* < 0.001), and SO (*p* = 0.002) in the mediolateral (ML) direction than in the anteroposterior (AP) direction. Except for the GM (*p* = 0.003), the correlation coefficient is higher in the AP direction than in the ML direction. For the leg dominance effects, the smaller correlation coefficient for the ST (*p* = 0.003) and the tendency for a smaller correlation coefficient for the GM (*p* = 0.015) exist for a non-dominant leg (ND) stance than for a dominant leg (DO) stance. For the surface stability effects, the tendency of the lower correlation coefficient for the RF (*p* = 0.020) and BF (*p* = 0.015) appears when balancing on an unstable surface (US) rather than when balancing on a stable surface (SS).

The interaction effects are observed in the specific pairs of factors and in the specific muscles. Specifically, the interactions of surface stability*sway direction are significant in the ST (*p* = 0.010), TA (*p* = 0.001), GM (*p* = 0.001), and SO (*p* < 0.001). Moreover, the interaction of sway direction*leg dominance is significant in the GM (*p* = 0.009).

### 3.3. Patterns of EMG–COP Correlation

Figure 3A shows the main positive contributions of the ST, BF, TA, PL, and SO in PC_1_, GM and SO in PC_2_, RF and BF in PC_3_, RF in PC_4_, ST and SO in PC_5_, BF and SO in PC_6_, and PL in PC_7_. In addition, the explained variances of individual PCs (PC_1–7_) are reported in the bracket, showing how much (in percent) individual EMG–COP correlations are involved in the total variances.

Figure 3B depicts a relationship between the first two patterns, PC_1_ and PC_2_, created for all factors. Only the sway direction factor (central panel) shows that PC_1_ can separate a difference in the muscle group involved in the AP and ML sways. The negative values of COP_ap_ indicate the inverse direction of muscle involvement compared to COP_ml_.

Table 3 represents the relative explained variance (rVAR_k_) of the individual principal components (PC_1–7_) tested by three factors: surface stability, sway direction, and leg dominance. The main results show that no significant differences are observed, but the tendency for differences exists in the specific patterns of two factors: surface stability (PC_2_, *p* = 0.023 and PC_5_, *p* = 0.013) and sway direction (PC_5_, *p* = 0.030 and PC_7_, *p* = 0.042). Moreover, only the significant interaction of sway direction*leg dominance is observed in PC_1_ (*p* = 0.008).

## 4. Discussion

The current study investigated the cross-correlation between lower-limb EMG activity and COP displacement (i.e., EMG–COP correlation) in physically active young adults performing unipedal balancing on stable and unstable surfaces. Three factors that might influence postural control—surface stability, sway direction, and leg dominance—were tested on two levels of EMG–COP correlations: individual muscles and multi-muscle patterns correlated to postural sway. The tested factor effects were expected to be observed in the specific muscles or in the specific multi-muscle patterns relevant to the unipedal balance task. However, the main results show that only the effects of sway direction and leg dominance are observed in the specific muscles.

Although several muscles of the stance leg are correlated with postural sway (i.e., COP displacements), the effects of sway direction and leg dominance observed in the specific muscles indicate their relevance to the specific cases. In this regard, two main points were discussed. First, when sway direction is considered, the increased correlations of the medial knee flexor (ST), foot invertor and evertor (TA and PL), and one-joint plantar flexor (SO) muscles with COP_ml_ could reflect their distinct roles in controlling unipedal stability on a small base of support in the frontal plane [14]. However, only the two-joint plantar flexor muscle (GM) is significantly more implicated in COP_ap_ than in COP_ml_, consistent with its anatomical action during anteroposterior sway [14]. Second, the reduced contribution of the medial knee flexor (ST) in the non-dominant leg indicates asymmetry in the hamstring muscle function responsible for unipedal postural stability. This point may be of interest, since asymmetry in neuromuscular control due to laterality effects has been considered one of the internal risks of sports-related knee injuries [23,24,25].

When considering the level of multi-muscle patterns, none of the examined factors influenced the patterns of lower-limb EMG activities associated with COP trajectories assessed through PCA. These findings might be influenced by differences in anatomical structures or practical functions across specific muscles [33], by different muscle recruitment patterns during the accomplishment of the same task [30], or by unexpected use or control of a movement component due to unusual instability events [38]. When focusing on the first main pattern (PC_1_), the positive correlations of the knee flexor (ST and BF), foot invertor (TA), and foot evertor (PL) muscles could reflect the slightly bent knee with the everted foot posture, which can be observed in the main movement component/synergy used to achieve unipedal equilibrium on stable [7] and multiaxial-unstable [8] surfaces. Additionally, when considering the second pattern (PC_2_), this pattern indicates the involvement of two plantar flexors (GM and SO), which are consistently correlated to the movement of the anteroposterior ankle strategy [14]. The current findings may serve as a pilot study for future work.

In terms of practical application, considering the asymmetry of lower-limb muscle activations involved in maintaining unipedal stability is of interest for assessing and training lower-limb performance, since the prevalence of several types of sports-related lower-limb injuries differs between the dominant and non-dominant legs [23,25,39,40]. Also, the mechanisms of lower-limb injury due to laterality effects vary by sport [41], but a training program that aims to reduce the inherent difference in neuromuscular control between the two legs may benefit sports-related injury prevention and rehabilitation.

### Limitations and Future Study

One limitation of the current study is that different types of EMG–COP correlations are not considered, e.g., positive and negative cross-correlation (Figure 2). This issue may require further investigation, since several muscles cooperatively control postural movements, possibly for distinct purposes (e.g., to drive or stabilize the movement) [14].

In addition, although lacking postural stability assessments, all participants are physically active young adults without neuromuscular impairments affecting their postural control ability. Regarding this point, analyzing the correlation between lower-limb myoelectric activity and postural stability in individuals with and without a history of injuries, e.g., ankle or knee injuries, is of interest for future research.

## 5. Conclusions

The current study investigated the temporal similarity in shape between lower-limb EMG activities and COP displacements (i.e., EMG–COP correlation) in physically active young adults during unipedal balancing on stable and multiaxial-unstable surfaces. The main findings show that only the effects of sway direction and leg dominance on EMG–COP correlations exist in the specific muscles, suggesting that the inherent difference in neuromuscular control of the specific muscles should be considered for assessing and training unipedal balance.

## Figures and Tables

**Figure 1 sports-10-00155-f001:**
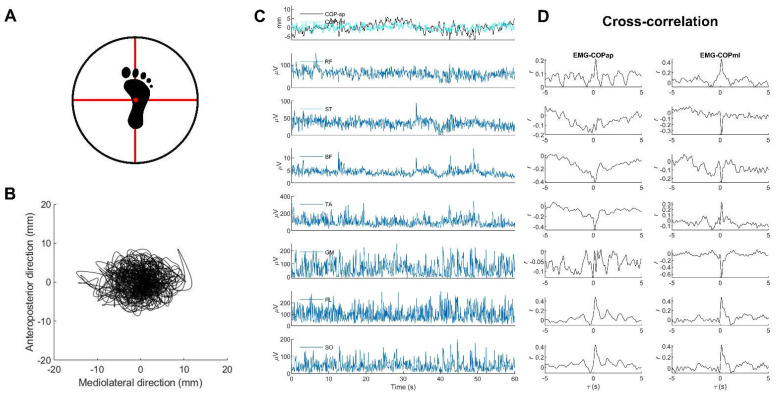
Examples of (**A**) the standardized foot position on a multiaxial-unstable surface, (**B**) statokinesigram, (**C**) COP stabilograms and EMG signals, and (**D**) corresponding cross-correlation plots analyzed from the entire 60-s data. Note: all EMG and COP data were derived from one participant during unipedal standing on the right leg on a multiaxial-unstable surface.

**Figure 2 sports-10-00155-f002:**
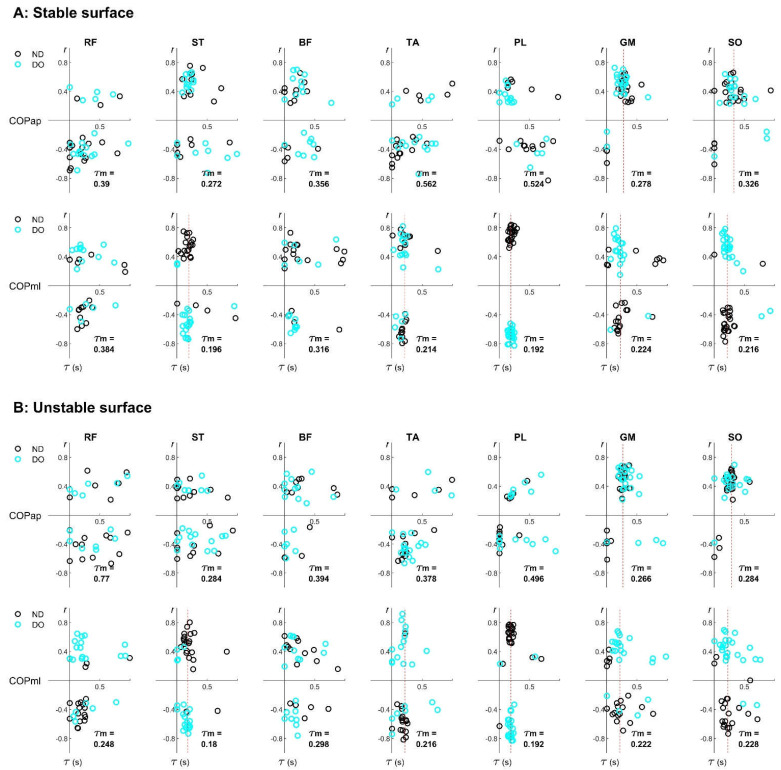
Examples of the subject-specific EMG–COP correlation of seven lower-limb muscles during unipedal balancing on (**A**) stable and (**B**) unstable surfaces. Note: each participant contributed two data points from the non-dominant (ND) and dominant (DO) legs, derived from the first subset of the whole 60-s data. A vertical red line was inserted to represent the alignment at the median time delay (τm).

**Figure 3 sports-10-00155-f003:**
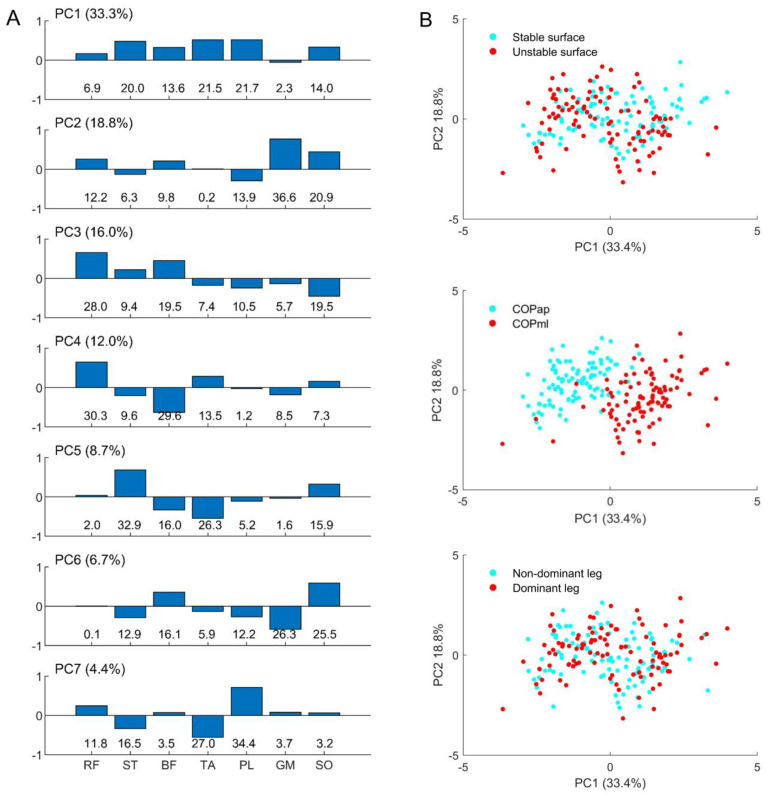
Representations of (**A**) the coefficient of individual principal components (PC_1–7_, explained variance in a bracket) with their contributions in percent (%, bottom row) to the eigenvector; and illustrative examples of (**B**) the relationship between PC_1_ and PC_2_ classified by individual factors: support surfaces (upper), sway direction (middle), and leg dominance (lower).

**Table 1 sports-10-00155-t001:** Characteristics of participants (mean ± SD; * *p* < 0.001).

	Total (*n* = 25)	Male (*n* = 14)	Female (*n* = 11)
Age (years)	25.6 ± 4.0	25.9 ± 2.9	25.3 ± 5.1
Weight (kg)	71.0 ± 11.5	77.0 ± 10.8	62.7 ± 5.2 *
Height (cm)	175.0 ± 8.3	180.1 ± 7.2	168.5 ± 3.9 *
Body mass index (kg/m^2^)	23.1 ± 2.7	23.9 ± 2.8	22.1 ± 2.3
Physical activity participation (hours/week)	8.4 ± 5.1	8.1 ± 5.5	8.8 ± 4.7

Note: the symbol * represents *p*-values that satisfy the Bonferroni-Holm criterion.

**Table 2 sports-10-00155-t002:** The z -transformed EMG–COP correlation coefficients (mean ± SD) tested on three factors: (A) surface stability, (B) sway direction, and (C) leg dominance.

A: Surface Stability					
Muscles	SS	US	*p*-Value	Effect size	Power
RF	0.40 ± 0.07	0.38 ± 0.07	0.020 ^#^	0.277	0.389
ST	0.47 ± 0.10	0.45 ± 0.10	0.164	0.192	0.214
BF	0.42 ± 0.08	0.40 ± 0.06	0.015 ^#^	0.342	0.545
TA	0.51 ± 0.09	0.49 ± 0.12	0.125	0.192	0.214
PL	0.58 ± 0.10	0.57 ± 0.13	0.621	0.073	0.072
GM	0.50 ± 0.09	0.50 ± 0.11	0.730	0.056	0.063
SO	0.50 ± 0.11	0.49 ± 0.11	0.585	0.093	0.087
**B: Sway Direction**					
**Muscles**	**AP**	**ML**	** *p* ** **-** **Value**	**Effect size**	**Power**
RF	0.40 ± 0.07	0.39 ± 0.07	0.232	0.159	0.161
ST	0.40 ± 0.07	0.52 ± 0.13	<0.001 *	−1.171	0.999
BF	0.40 ± 0.06	0.42 ± 0.09	0.139	−0.233	0.292
TA	0.42 ± 0.08	0.58 ± 0.13	<0.001 *	−1.503	1
PL	0.37 ± 0.06	0.78 ± 0.17	<0.001 *	−3.208	1
GM	0.54 ± 0.10	0.47 ± 0.10	0.003 *	0.726	0.992
SO	0.46 ± 0.09	0.54 ± 0.12	0.002 *	−0.783	0.997
**C: Leg Dominance**					
**Muscles**	**ND**	**DO**	** *p* ** **-** **Value**	**Effect size**	**Power**
RF	0.39 ± 0.08	0.40 ± 0.07	0.275	−0.156	0.157
ST	0.44 ± 0.10	0.48 ± 0.11	0.003 *	−0.462	0.799
BF	0.41 ± 0.07	0.41 ± 0.08	0.708	−0.066	0.068
TA	0.50 ± 0.11	0.50 ± 0.10	0.838	0.036	0.055
PL	0.58 ± 0.11	0.57 ± 0.12	0.765	0.041	0.057
GM	0.49 ± 0.09	0.52 ± 0.10	0.015 ^#^	−0.280	0.396
SO	0.50 ± 0.11	0.50 ± 0.11	0.661	−0.048	0.060

Note: the symbol ^#^ represents *p*-values smaller than 0.05, and the symbol * represents *p*-values that satisfy the Bonferroni-Holm criterion.

**Table 3 sports-10-00155-t003:** The relative explained variance (rVAR_k_; mean ± SD) of individual principal components (PC_1–7_) tested on three factors: (A) surface stability, (B) sway direction, and (C) leg dominance.

A: Surface Stability					
rVAR_k_	SS	US	*p*-Value	Effect size	Power
1	24.7 ± 14.0	24.4 ± 14.3	0.839	0.026	0.053
2	15.2 ± 9.1	19.5 ± 13.3	0.023 ^#^	−0.379	0.632
3	18.2 ± 12.1	14.4 ± 10.1	0.061	0.345	0.552
4	13.9 ± 9.2	12.0 ± 9.0	0.177	0.211	0.248
5	9.9 ± 6.8	12.3 ± 7.5	0.013 ^#^	−0.342	0.545
6	10.3 ± 6.7	10.1 ± 6.7	0.893	0.020	0.052
7	7.8 ± 6.7	7.3 ± 6.5	0.584	0.077	0.075
**B: Sway Direction**					
**rVAR_k_**	**AP**	**ML**	** *p* ** **-** **Value**	**Effect size**	**Power**
1	26.8 ± 16.0	22.3 ± 12.3	0.129	0.317	0.484
2	17.9 ± 10.6	16.8 ± 11.7	0.574	0.101	0.094
3	17.5 ± 12.0	15.1 ± 10.1	0.224	0.221	0.268
4	12.1 ± 8.9	13.8 ± 9.3	0.234	−0.187	0.205
5	9.8 ± 6.7	12.4 ± 7.6	0.030 ^#^	−0.372	0.615
6	9.4 ± 6.4	11.0 ± 7.0	0.128	−0.237	0.300
7	6.5 ± 5.2	8.6 ± 8.0	0.042 ^#^	−0.320	0.491
**C: Leg Dominance**					
**rVAR_k_**	**ND**	**DO**	** *p* ** **-** **Value**	**Effect size**	**Power**
1	23.3 ± 14.1	25.8 ± 14.2	0.174	−0.172	0.181
2	17.6 ± 11.0	17.0 ± 11.3	0.749	0.052	0.061
3	16.0 ± 10.8	16.6 ± 11.4	0.711	−0.048	0.060
4	13.8 ± 9.6	12.1 ± 8.6	0.166	0.186	0.204
5	11.5 ± 7.1	10.7 ± 7.3	0.510	0.120	0.112
6	10.0 ± 6.8	10.4 ± 6.6	0.636	−0.055	0.063
7	7.7 ± 7.0	7.5 ± 6.2	0.833	0.030	0.054

Note: the symbol ^#^ represents *p*-values smaller than 0.05; however, none of the differences satisfied the Bonferroni-Holm criterion.

## Data Availability

Not applicable.

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
