# Peer review of "Modulation of Lower-Limb Muscle Activity in Maintaining Unipedal Balance According to Surface Stability, Sway Direction, and Leg Dominance"

_sports, 2022, doi:10.3390/sports10100155_

Round 1

Reviewer 1 Report

Congratulations on the article, you can see the work that has gone into its presentation. In general, the article is very well written and has some very clear figures. 

There are some correctable errors and editing of the article above all, although some content errors have also been detected, even so, congratulations, it has a great quality.

Title

The title is correctly described and refers to the implications of the article and is well written.

Abstract 

The abstract is of good quality and has a clear explanation of its components, but the parts of the paper could be separated, as it is not clear, for example, where the theoretical framework ends or the methodology begins for a reader who has not yet read the full text.

In addition, the objective of the work should be clearly and concretely stated at this point of the work and in this summary, it is not specifically differentiated.

Keywords

An excessive number of keywords has been selected, which may confuse the reader. 4 or 6 keywords would be enough, and not all of them have the same weight in this research article. Select only the most relevant ones, as all of them are related to the present article.

1. Introduction 

Reference 19 and 20 should be interchanged numerically for the correct appearance in the text.

The objective of the research is well defined and, moreover, it is well supported by the hypotheses launched, but it is described twice, and only once in the last paragraph would be enough.

It is rich in bibliography and well written from the general to the specific.

2. Materials and methods 

2.1. Participants 

This section of the paper is well written in form and content. But it could be added that the selected subjects are physically active within the text, as in the description of the objective, although the table clearly shows the average hours of sport per week and their standard deviations.

2.2. Equipment

The equipment used to extract the EMG and COP data and its technical characteristics are clear, as well as the placement of the electrodes and the manual used.

The units used in both cases, for EMG and COP, can be added so that the reader has a quicker understanding when coming to figure 1.

2.3. Experimental procedures

All the content of this section is correctly developed, but it should be added how the test is developed in unstable conditions, giving the reader a better understanding of the experiment and also, the type of rest to follow, in case an active or passive rest is established, or at the choice of the participant. 

2.4. Data analysis

2.4.1. Pre-processing of EMG and COP data 

The pre-processing is clear and well done for the correct visualisation and treatment of the data, and well supported by the literature. Simply, as an edit, it could be written in such a way that the paragraph ends and then the formulas are shown, it would be more visually pleasing that way. 

2.4.2. Calculation of EMG-COP correlation

This section is well defined and supported by scientific literature. 

I would like to emphasise the accuracy of the image shown, which contains the information on the typology of the data collected graphically, and the location seems to me to be correct, as it includes all the sections dealt with in the previous table. Even so, image c of the table should be better cropped so that the units can be seen in all the graphs presented.

2.4.3. Analysis of EMG-COP correlation patterns

The explanation of the procedure is clear and bibliographically supported.

In the last paragraph, different acronyms are used to refer to the same fact, rVARk and rVAR.

2.5.     Statistical analysis 

This section is well developed and is a good end to the section on materials and methods.

3.        Results

Well presented data extracted from the research and clarity in the images used and explained.

3.1. Individual EMG-COP correlation pairs 

When talking about the correlation and that it is moderate to high, we are talking in a general way, but we can refine it a bit more, showing the maximum data obtained in the peroneus longus, for example.

3.2. EMG-COP correlation patterns

In figure 3A it is found that PC5 is wrongly worded according to the graphs shown and should refer to SO and not to GM as it is in the text right now.

4.         Discussion

This point is correctly worded and supported by the bibliography, in the last paragraph, the last sentence should be added in a point on future research.

4.1. Limitations 

The limitations of the study are well developed, I would only add that the population is young and physically active. In addition, although some future research is presented, a paragraph could be added with the lines of future research that the author believes to be the most appropriate, grouped in one place. 

5.         Conclusion

In the conclusion it should be clear which hypothesis has been fulfilled after the study has been carried out and we should not add new data, as is expressed with the semitendinosus which should be in the discussion section.

Author Response

Response to Reviewer 1 Comments

Point 1: Congratulations on the article, you can see the work that has gone into its presentation. In general, the article is very well written and has some very clear figures. 

There are some correctable errors and editing of the article above all, although some content errors have also been detected, even so, congratulations, it has a great quality.

Title: The title is correctly described and refers to the implications of the article and is well written.

Response 1: Thank you very much for your valuable time in carefully reviewing the manuscript and providing constructive comments and valuable suggestions. I appreciate all the suggestions and applied them to improve the study. Please find all the changes in the revised manuscript and briefly in the point-by-point responses below.

Point 2:

Abstract: The abstract is of good quality and has a clear explanation of its components, but the parts of the paper could be separated, as it is not clear, for example, where the theoretical framework ends, or the methodology begins for a reader who has not yet read the full text.

In addition, the objective of the work should be clearly and concretely stated at this point of the work, and in this summary, it is not specifically differentiated.

Response 2: Thank you for the comments. In the revised manuscript, I rewrote the Abstract by reorganizing the information of each sub-content to make it more precise than the old one.

Point 3:

Keywords: An excessive number of keywords has been selected, which may confuse the reader. 4 or 6 keywords would be enough, and not all of them have the same weight in this research article. Select only the most relevant ones, as all of them are related to the present article.

Response 3: I agree with the suggestion and reduced the number of keywords in the revised manuscript by selecting only five keywords (neuromuscular control, postural control, single-leg stance, electromyography (EMG), and center of pressure (COP)).

Point 4:

  1. Introduction: Reference 19 and 20 should be interchanged numerically for the correct appearance in the text. The objective of the research is well defined and, moreover, it is well supported by the hypotheses launched, but it is described twice, and only once in the last paragraph would be enough. It is rich in bibliography and well written from the general to the specific.

Response 4: Thank you for the suggestions. I rewrote the Introduction in the revised manuscript by removing the duplicate information.

Point 5:

  1. Materials and methods 

2.1. Participants: This section of the paper is well-written in form and content. But it could be added that the selected subjects are physically active within the text, as in the description of the objective, although the table clearly shows the average hours of sport per week and their standard deviations.

Response 5: In the revised manuscript, I already replaced the word “healthy” with “physically active”.

Point 6:

2.2. Equipment: The equipment used to extract the EMG and COP data and its technical characteristics are clear, as well as the placement of the electrodes and the manual used.

The units used in both cases, for EMG and COP, can be added so that the reader has a quicker understanding when coming to figure 1.

Response 6: I added the units of EMG and COP signals accordingly.

Point 7:

2.3. Experimental procedures: All the content of this section is correctly developed, but it should be added how the test is developed in unstable conditions, giving the reader a better understanding of the experiment and also, the type of rest to follow, in case an active or passive rest is established, or at the choice of the participant. 

Response 7: I already added more information about the unstable conditions in the revised manuscript.

Point 8:

2.4. Data analysis

2.4.1. Pre-processing of EMG and COP data: The pre-processing is clear and well done for the correct visualisation and treatment of the data, and well supported by the literature. Simply, as an edit, it could be written in such a way that the paragraph ends and then the formulas are shown, it would be more visually pleasing that way. 

Response 8: Thank you for the suggestion. I already improved the mentioned sentence in the revised manuscript by adding the formulas at the end.

Point 9:

2.4.2. Calculation of EMG-COP correlation

This section is well defined and supported by scientific literature. 

I would like to emphasise the accuracy of the image shown, which contains the information on the typology of the data collected graphically, and the location seems to me to be correct, as it includes all the sections dealt with in the previous table. Even so, image c of the table should be better cropped so that the units can be seen in all the graphs presented.

Response 9: I created the new Figure 1 with high resolution and saw all the signal units.

Point 10:

 2.4.3. Analysis of EMG-COP correlation patterns

The explanation of the procedure is clear and bibliographically supported.

In the last paragraph, different acronyms are used to refer to the same fact, rVARk and rVAR.

Response 10: thank you for pointing out the mistake. I corrected the mistake already.

Point 11:

2.5.     Statistical analysis 

This section is well developed and is a good end to the section on materials and methods.

Response 11: Thank you.

Point 12:

  1. Results

Well-presented data extracted from the research and clarity in the images used and explained.

3.1. Individual EMG-COP correlation pairs 

When talking about the correlation and that it is moderate to high, we are talking in a general way, but we can refine it a bit more, showing the maximum data obtained in the peroneus longus, for example.

Response 12: I added more information about the results shown in this sub-section by adding the information about the highest value observed in the peroneus longus, as suggested already.

Point 13:

 3.2. EMG-COP correlation patterns

In figure 3A it is found that PC5 is wrongly worded according to the graphs shown and should refer to SO and not to GM as it is in the text right now.

Response 13: Thank you for pointing out the mistake, I corrected that information in the revised manuscript already.

Point 14:

  1. Discussion

This point is correctly worded and supported by the bibliography, in the last paragraph, the last sentence should be added in a point on future research.

Response 14: I added the sentence to suggest for future study in the last section of the Discussion already.

Point 15:

4.1. Limitations 

The limitations of the study are well developed, I would only add that the population is young and physically active. In addition, although some future research is presented, a paragraph could be added with the lines of future research that the author believes to be the most appropriate, grouped in one place. 

Response 15: Thank you for the suggestion, I did use the word “physically active” in the revised manuscript accordingly.

Point 16:

  1. Conclusion

In the conclusion, it should be clear which hypothesis has been fulfilled after the study has been carried out and we should not add new data, as is expressed with the semitendinosus, which should be in the discussion section.

Response 16: I rewrote the Conclusion based on the suggestion in the revised manuscript. Thank you very much again for providing valuable suggestions to improve the manuscript.

Reviewer 2 Report

Thank you for presenting a well-written, designed, and considered article. It was a pleasure to read. My only concern is the limited discussion regarding the real-world meaningfulness of your results. You have, in my opinion, over-emphasised the 'potential' role of ST in injury.

Author Response

Response to Reviewer 2 Comments

Point 1: Thank you for presenting a well-written, designed, and considered article. It was a pleasure to read. My only concern is the limited discussion regarding the real-world meaningfulness of your results. You have, in my opinion, over-emphasized the 'potential' role of ST in injury.

Response 1: Thank you for your valuable time on this study. I appreciate all the helpful comments and especially for pointing out the issue that should be addressed. I agree with the comments. I rewrote this part in the revised manuscript to avoid making an over-emphasis.

Reviewer 3 Report

Thank you for the possibility to review the article entitled “Modulation of lower-limb muscle activity in maintaining  unipedal balance according to surface stability, sway direction, and leg dominance”. The article was generally well written, the argumentation is sound, and the paper well structured. In my opinion, the statistical approach is suitable to test the hypotheses.

The opening paragraph is too vague and does really lay a good foundation for the context of the study.

The goal of the study needs to be properly highlighted and justified. Instead of setting their aim in the frame of a simple question (i.e., The current study investigated unipedal postural control on stable and unstable surfaces by determining how much EMG activity of the lower-limb muscles correlated with  COP displacements), I would recommend that the authors attempt to present the key objectives of their study with regards to what is presently known (i.e. literature), thus highlighting the added value of the article.

The authors should definitely elaborate on the hypothesis as they are not sufficiently backed with theoretical considerations.

I encourage authors describe how particpants were recruited. Were they male? Female?

A priori power analysis? Did the authors calculate an a priori power analysis to determine the sample size?  

I encourage authors to provide partial eta squared (ƞp2) for ANOVA and Cohen’s d for post-hoc comparisons.

It would be appreciated if the authors could give more details about practical implications of the study

Author Response

Response to Reviewer 3 Comments

Point 1: Thank you for the possibility to review the article entitled “Modulation of lower-limb muscle activity in maintaining unipedal balance according to surface stability, sway direction, and leg dominance”. The article was generally well written, the argumentation is sound, and the paper is well structured. In my opinion, the statistical approach is suitable to test the hypotheses.

Response 1: Thank you very much for your valuable time on this study. I appreciate all the valuable comments and suggestions. In the revised manuscript, I applied all the comments and suggestions to improve it and hope it might be clearer than the old one. Please find my brief changes in the responses below and the full details in the manuscript.

Point 2: The opening paragraph is too vague and does really lay a good foundation for the context of the study.

Response 2: Thank you for your comments. I tried to add more information and rewrote the opening paragraph in the revised manuscript.

Point 3: The goal of the study needs to be properly highlighted and justified. Instead of setting their aim in the frame of a simple question (i.e., The current study investigated unipedal postural control on stable and unstable surfaces by determining how much EMG activity of the lower-limb muscles correlated with COP displacements), I would recommend that the authors attempt to present the key objectives of their study with regards to what is presently known (i.e., literature), thus highlighting the added value of the article.

The authors should definitely elaborate on the hypothesis as they are not sufficiently backed with theoretical considerations.

Response 3: I rewrote the goals and hypotheses of the study based on the suggestions by adding more information.

Point 4: I encourage authors to describe participants who were recruited. Were they male? Female?

A priori power analysis? Did the authors calculate an a priori power analysis to determine the sample size? 

Response 4: I added information about the sexes. Although I did not perform the Priori power analysis, the number of participants included in this study is more than the number of participants reported in similar studies (e.g., 15 participants in Lemos et al. (2015), 13 participants in Lemos et al. (2014), 9 participants in Croft et al. (2008) and 12 participants in Sozzi et al. (2013)). However, thank you for pointing out the important process that should be considered. I will apply a priori power analysis to determine the sample size for the future study.

Point 5: I encourage authors to provide partial eta squared (ƞp2) for ANOVA and Cohen’s d for post-hoc comparisons.

Response 5: Thank you very much. I used Cohen’s d for post hoc comparisons according to the suggestion.

Point 6: It would be appreciated if the authors could give more details about the practical implications of the study

Response 6: I added one more paragraph regarding the practical implication in the discussion according to the suggestion. Thank you very much again for providing valuable suggestions to improve the manuscript.

Round 2

Reviewer 3 Report

The paper is now suitable for publication